# An international consensus for mitigation of the detrimental effects of the COVID-19 pandemic on laparoscopic training

Marina Yiasemidou[1,2]*, Annabel Howitt[3], Judith Long[4], Peter Sedman[4], Damian Garcia-Olmo[5], Hector Guadalajara[6], Ben Van Cleynenbreugel[7], Dhananjaya Sharma[8], Shekhar Chandra Biyani[9], Bijendra Patel[10], Wayne Lam[11], Athur Harikrishnan[12], Juan Gómez Rivas[13], Jonathan Robinson[14], Tiago Manuel Ribeiro de Oliveira[15], Gabriel Escalona Vivas[16], Rafael Sanchez-Salas[17], Rafael Tourinho-Barbosa[18], Ian Chetter[4]

**1** NIHR Academic Clinical Lecturer, University of Hull, Hull, United Kingdom, **2** ST8 Colorectal Surgery, Bradford Teaching Hospitals, Bradford, United Kingdom, **3** General Surgery, Bradford Teaching Hospitals, Bradford, United Kingdom, **4** Hull University Teaching Hospitals, Hull, United Kingdom, **5** Chief of Surgery Department at Fundacion Jimenez Díaz University Hospital (Universidad Autonoma de Madrid), Madrid, Spain, **6** University Hospital Fundación Jimenez Díaz and Associate Professor at "Universidad Autonoma de Madrid", Madrid, Spain, **7** University Hospitals of Leuven and a Guest Lecturer at the Catholic University of Leuven, Leuven, Belgium, **8** NSCB Government Medical College Jabalpur (MP), Jabalpur, India, **9** Leeds Teaching Hospitals and Honorary Senior Clinical Lecturer, University of Leeds, Leeds, United Kingdom, **10** Barts Health NHS Trust and University College London Hospitals NHS Foundation Trust and Professor of Surgery, University of London, London, United Kingdom, **11** Department of Surgery, Faculty of Medicine, Queen Mary Hospital, University of Hong Kong, Hong Kong, China, **12** Sheffield Teaching Hospitals, Sheffield, United Kingdom, **13** University Hospital La Paz, Madrid, Spain, **14** Bradford Teaching Hospitals, Bradford, United Kingdom, **15** Consultant Urologist, Lisbon, Portugal, **16** Hospital Sótero del Río and researcher at the Simulation and Experimental Surgery Center at the Pontificia Universidad Católica in Santiago, Santiago, Chile, **17** Department of Urology L'Institut Mutualiste Montsouris Université, Paris Descartes, Paris, France, **18** Department of Urology Hospital Cardio Pulmonar–Oncologia D'Or and Núcleo de Oncologia da Bahia, Faculdade de Medicina do ABC, Salvador, Bahia, Brazil

* marinayiasemidou@gmail.com

## Abstract

### Aim

Achieve an international consensus on how to recover lost training opportunities. The results of this study will help inform future EAES guidelines about the recovery of surgical training before and after the pandemic.

### Background

A global survey conducted by our team demonstrated significant disruption in surgical training during the COVID-19 pandemic. This was wide-spread and affected all healthcare systems (whether insurance based or funded by public funds) in all participating countries. Thematic analysis revealed the factors perceived by trainees as barriers to training and gave birth to four-point framework of recovery. These are recommendations that can be easily achieved in any country, with minimal resources. Their implementation, however, relies heavily on the active participation and leadership by trainers. Based on the results of the global trainee survey, the authors would like to conduct a Delphi-style survey, addressed to trainers on this occasion, to establish a pragmatic step-by-step approach to improve training during and after the pandemic.

**Data Availability Statement:** No datasets were generated or analysed during the current study. All relevant data from this study will be made available upon study completion.

**Funding:** The project is funded by EAES - European Association Endoscopic Surgery. The funders had no role in study design, data collection and analysis, decision to publish, or preparation of the manuscript.

**Competing interests:** The authors have declared that no competing interests exist.

## Methods

This will be a mixed qualitative and quantitative study. Semi-structured interviews will be performed with laparoscopic trainers. These will be transcribed and thematic analysis will be applied. A questionnaire will then be proposed; this will be based on both the results of the semi structured interviews and of the global trainee survey. The questionnaire will then be validated by the steering committee of this group (achieve consensus of >80%). After validation, the questionnaire will be disseminated to trainers across the globe. Participants will be asked to consent to participate in further cycles of the Delphi process until more than 80% agreement is achieved.

## Results

This study will result in a pragmatic framework for continuation of surgical training during and after the pandemic (with special focus on minimally invasive surgery training).

## 1. The research team

### a. Principle applicant

Miss Marina Yiasemidou, MBBS, MSc, MRCS, PGCer, PhD
   National Institute of Health Research (NIHR) Academic Clinical Lecturer in General Surgery,
   University of Hull
   Email: marinayiasemidou@gmail.com
   M.Yiasemidou@nhs.net
   Tel: +44 (0) 7975531067
   I designed this project and will be part of the steering committee overseeing and ensuring its successful completion. Besides co-ordinating and facilitating communication between the members of the steering committee, the co-applicants and other experts, I will have an active role in the thematic analysis of the transcripts of the semi-structured interviews and the interim and final analysis of the Delphi-style survey. I will lead the write up and dissemination of the outcomes of this study.

### b. Co-investigators (title, contact details, roles in the project)

The role of the surgical educationalists is explained in detail in the methodology section. Amongst them are the members of the steering committee, who will be formulating the questions to be asked during semi-structured interviews and will act as the 'facilitator' during the Delphi-style survey. The remaining surgical educationalists can be interviewed and can participate in the Delphi-style survey. They will also be expected to recruit other specialists for the Delphi-style survey.
   **(i) Surgical educationalists.**

1. **Peter Sedman,**
   Consultant Upper GI surgeon Hull University Teaching Hospitals, Hull, UK
   peter.sedman@hey.nhs.uk

2. **Mr Jonathan Robinson**
   Consultant Colorectal Surgeon, Bradford Teaching Hospitals Foundation Trust, Bradford, UK
   Jonathan.Robinson@bthft.nhs.uk

3. **Professor Ian Chetter**
Professor of Surgery, Hull and York Medical School and Consultant Surgeon Hull University Teaching Hospitals, Hull, UK
Ian.Chetter@hey.nhs.uk

4. **Professor Damian Garcia-Olmo,**
Professor of Surgery and Chief of Surgery Department at Fundacion Jimenez Díaz University Hospital (Universidad Autonoma de Madrid), Spain
damian.garcia@uam.es

5. **Hector Guadalajara,**
Colorectal surgeon, Chairperson for surgery at University Hospital Fundación Jimenez Díaz and Associate Professor at "Universidad Autonoma de Madrid", Spain
h.guadalajara@quironsalud.es

6. **Ben Van Cleynenbreugel**
Consultant urologist at the University Hospitals of Leuven and a guest Lecturer at the Catholic University of Leuven, Belgium
Ben.Vancleynenbreugel@uzleuven.be

7. **Professor Dhananjaya Sharma,**
Professor of Surgery and Head of Department of Surgery NSCB Government Medical College Jabalpur (MP), India
dhanshar@gmail.com

8. **Shekhar Chandra Biyani**
Consultant Urologist, Leeds Teaching Hospitals and Honorary Senior Clinical Lecturer, University of Leeds, Leeds, UK
shekharbiyani@hotmail.com

9. **Professor Bijendra Patel**
Consultant Upper GI & Laparoscopic Surgeon at Barts Health NHS Trust and University College London Hospitals NHS Foundation Trust and Professor of Surgery, University of London, UK
b.patel@qmul.ac.uk

10. **Wayne Lam**
Clinical Assistant Professor in Urology, Department of Surgery,
Faculty of Medicine, Queen Mary Hospital, University of Hong Kong
waynelam.urology@gmail.com

11. **Arthur Harikrishnan**
Consultant Colorectal Surgeon, Sheffield Teaching Hospitals, Sheffield, UK
harikrishnan@doctors.org.uk

12. **Juan Gómez Rivas**
Consultant urologist, University Hospital *La Paz*, Madrid, Spain
juangomezr@gmail.com

13. **Tiago Manuel Ribeiro de Oliveira**
Consultant Urologist, Portugal
tiagoribeirooliveira@sapo.pt

14. **Gabriel Escalona Vivas**
General surgeon at Hospital Sótero del Río and researcher at the Simulation and Experimental Surgery Center at the Pontificia Universidad Católica in Santiago, Chile.
gabrielescalonav@gmail.com

15. **Rafael Sanchez-Salas**
Attending Surgeon, Department of Urology L'Institut Mutualiste Montsouris Université, Paris Descartes, Paris, France.
raersas@gmail.com

16. **Dr. Rafael Tourinho-Barbosa,**
Attending Surgeon, Department of UrologyHospital Cardio Pulmonar–Oncologia D'Or and Núcleo de Oncologia da Bahia, Faculdade de Medicina do ABC, Salvador, Bahia, Brazil.
rafael.tourinho@hotmail.com

17. **Annabel Howitt**
Clinical fellow, General Surgery
Bradford Teaching Hospitals Foundation Trust, Bradford, UK
annabelcj@hotmail.co.uk

18. **Judith Long**
Research Manager Academic Vascular Surgical Unit, Hull University Teaching Hospital NHS Trust, Hull, UK
judith.Long@hey.nhs.uk

## 2. Duration of the project–Gantt chart

| | May21 | | Jul'21 | | Sep'21 | | Nov'21 | | | | | | Apr'22 | | Jun'22 |
|---|---|---|---|---|---|---|---|---|---|---|---|---|---|---|---|
| Semi-structured interviews | ■ | ■ | | | | | | | | | | | | | |
| Preparation of Delphi survey | | | ■ | ■ | | | | | | | | | | | |
| Validation of Survey | | | | | ■ | ■ | | | | | | | | | |
| Delphi cycles | | | | | | | ■ | ■ | ■ | ■ | ■ | | | | |
| Data collection | | | | | | | | | | | | | ■ | ■ | |
| Dissemination | | | | | | | | | | | | | | | ■ |

As seen in the Gantt chart above, the project is expected to take fourteen months overall. The semi-structured interviews with take place within two months. The transcripts of the interviews will be analysed through thematic analysis and then, in combination with the global survey results, will be used to prepare the Delphi style survey. After a consensus is reached within the steering committee for the project, the survey will be validated by a small group of surgical educational experts within the next four months. The survey will then be released and several cycles will be undertaken until >80% consensus is achieved; a process expected to be completed within five months. Data collection will be done simultaneously to the release of the Delphi survey and will be completed two months after the last cycle of the survey. Dissemination will be done within a month after that.

## 3. Hypothesis / aims

### a. Aim

**Primary Aim: Establish an international consensus on how to recover lost training opportunities during and after the COVID-19 pandemic.**

**Secondary Aim: Help inform future EAES guidelines about the recovery of surgical training before and after the pandemic**.

Our group has completed a global trainee survey assessing the impact of the COVID-19 pandemic on surgical training. The results showed major disruption in all aspects of training, with a considerable impact on training in the operating theatre. Minimally invasive surgery was particularly affected due to being an aerosol generating procedure [1] with early reports suggesting it should be avoided [2, 3]. The collective responses generated a four pillar recovery framework which heavily relied on the engagement of and guidance from trainers. The aim of the proposed global Delphi style survey—aimed at trainers this time—is to assess how the targets set by trainees for the recovery of surgical training can be best achieved.

Any recovery plan lacking participation from both trainees and trainers is destined to fail. Through our team's previous global survey, we obtained the opinion of the trainees and we now wish to add further validity to the process by achieving an international expert (i.e. trainer) consensus on how to mitigate the negative effects of the COVID-19 pandemic on surgical training.

Recovery steps should be pragmatic and feasible in all healthcare systems despite of financial or other constrains.

## 4. Background and state of the art, including relevant bibliography

Since the declaration of the SARS-CoV-2 global pandemic on the 11[th] March [4], unprecedented measures were introduced to reduce the population exposure to the virus [5]. This brought significant changes in the delivery of surgical services. Semi-urgent and elective surgery, as well as endoscopy, were discontinued after relevant recommendations from professional bodies [6–12]. Surgical courses [13, 14], examinations, conferences and training rotations were cancelled [15].

In an attempt to minimise staff and patient exposure, many centres adopted a consultant only operating policy [16]. In a response to the high rates of adverse events, reported during the early days of the pandemic, common surgical conditions that would have been treated operatively priorly, have been managed conservatively [17–20]. This reduced operative training opportunities even further. Lack of face-to-face outpatient clinics also had a negative impact on training [15].

Moreover, whilst a significant amount of both elective and emergency work prior to the outbreak, consisted of laparoscopic and robotic surgery; surgical services around the globe had to adjust to emerging reports that minimally invasive surgery increases the exposure of the theatre team to aerosolised virus particles [21]. This is mostly attributed to the high-pressure insufflation process for establishment of pneumoperitoneum as well as the release of such particles during the deflation of the pneumoperitoneum [22, 23]. Due to minimally invasive surgery being identified as an aerosol generating procedure [1], it was considered potentially high risk for COVID-19 viral transmission.

As a result, several international and national surgical authorities have issued guidelines on the use of minimally invasive surgery during the COVID-19 pandemic; most of which recommended that minimally invasive surgery is used with caution [2, 3]. In an attempt to protect theatre staff from exposure, some centres discontinued minimally invasive surgery during the early days of the pandemic. Inevitably, this may have caused a reduction in the training opportunities associated with minimally invasive techniques.

Our group conducted a global survey which provided intelligence regarding the direct and indirect impact of the pandemic on surgical education and training and explored methods that might mitigate negative impacts on training, during and after the pandemic [24].

## 4.1. Direct and indirect impact of COVID-19 on training

A total of 608 responses from thirty-four countries were received (Table 1). The majority of responders reported significant disruption or complete discontinuation of all aspects of surgical training. The impact particularly affected conferences (525/608, 86%) and hands-on courses (such as simulation) (517/608, 85%). Out-patient clinic training (462/608, 76%), operative experience (483/608, 79%), endoscopy/cystoscopy (379/608, 62%), regional teaching (428/608, 70%) and training relating to in-patient care (268/608, 44%) were also greatly affected (Fig 1).

Training in the operating theatre appeared to be severely compromised, especially in Europe. Specifically, 89% of responders from Europe reported that their operating theatre training is affected to a great degree or they had no relevant training during the pandemic.

In addition to the direct impact on surgical training, there were several indirect consequences reported by trainees. The most commonly reported was interruption to career progression (54/202, 27%). Discontinuation of surgical rotations, exam cancellation and alteration of the recruitment processes (no face-to-face interviews due to social distancing rules), were some of the reasons cited by trainees for hindrance of career progression.

Focus on emergency care, deprived trainees from the small number of on-going elective activities (43/202, 21%). Redeployment to other specialties or new roles (e.g. ward based duties) resulted in disruption of the clinical team's coherence and mentoring by senior surgeons (34/202, 17%).

**Table 1. Demographics of responders.**

| *Gender* | |
|---|---|
| Male | 379 (62%) |
| Female | 227 (38%) |
| Other | 2 (0%) |
| *Countries* | |
| UK | 337 (55%) |
| Australia | 48 (8%) |
| Spain | 39 (6%) |
| China | 29 (5%) |
| India | 28 (5%) |
| Belgium | 24 (4%) |
| Hong Kong | 24 (4%) |
| Greece | 14 (2%) |
| Other | 65 (11%) |
| *Specialty* | |
| General surgery | 202 (33%) |
| Urology | 163 (27%) |
| Trauma and Orthopaedics | 67(11%) |
| Oral Maxillofacial surgery | 43 (7%) |
| Vascular surgery | 29 (5%) |
| Obstetrics and Gynaecology | 22 (4%) |
| Ear Nose Throat surgery | 18 (3%) |
| Plastics surgery | 13 (2%) |
| Other | 51 (8%) |
| *Surgical experience* | |
| <3 years | 198 (33%) |
| ≥3 years | 410 (67%) |

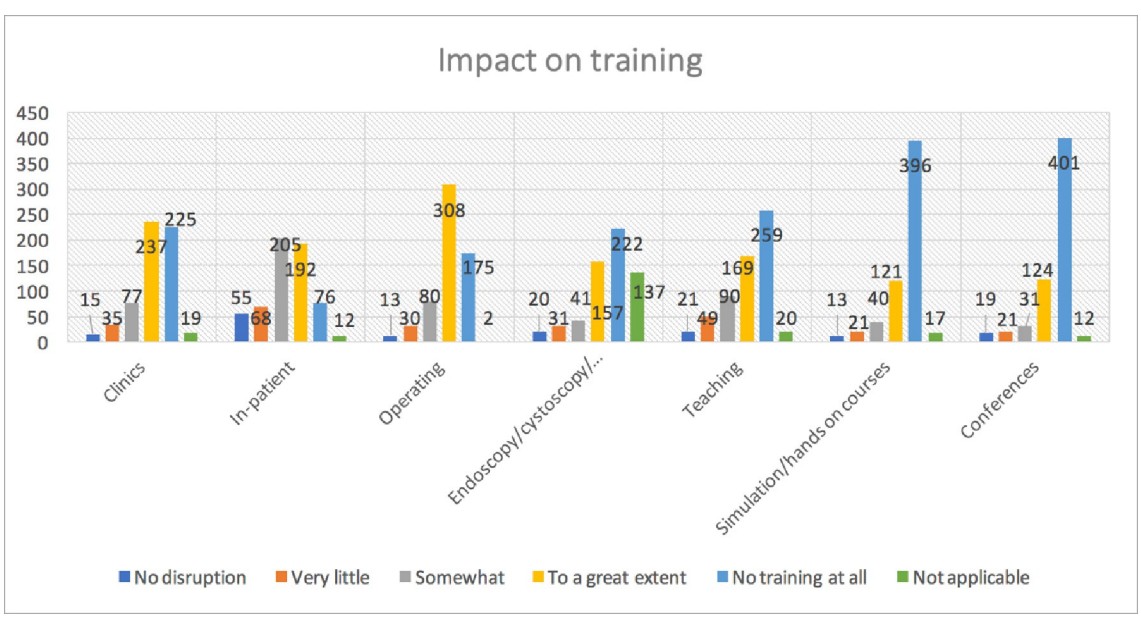

**Fig 1. Impact on training.**

Furthermore, a noticeable reduction in the number of emergency admissions during the first wave of COVID-19 reduced exposure and training opportunities in acute surgical patient management (10/202, 5%). Often a "minimal theatre staff" policy was applied, aiming to minimise the exposure of healthcare staff to aerosol producing procedures such as intubation and establishment of pneumoperitoneum for laparoscopic surgery. This restricted theatre access for trainees, especially diminishing opportunities to train in minimally invasive surgery.

## 4.2. Factors affecting provision of training during the pandemic

The trainees responding to our survey gave their perspectives about the factors affecting the provision of training during the pandemic. Lack of guidance from local or national training authorities (37%), reduced elective and emergency caseload and a "consultant only operating" policy (18%), limited access to appropriate equipment, including lack of IT (Information Technology) equipment, technical troubleshooting with internet connections and lack of simulators (65/373, 17%) were some of the most commonly identified reasons for hindering the provision of training during the COVID-19 pandemic. Less popular reasons included focus on service provision (38/373, 10%) discontinuation of educational activities due to social distancing (19/373, 5%) and lack of communication and coordination between training and other authorities (15/373, 4%).

New educational resources were rapidly introduced to counteract the detrimental effects on training; these included: webinars (359/608, 59%), online educational videos (234/608, 38%), virtual reality resources (34/608, 5.5%) and online learning quizzes (64/608, 10.5%) (Other less frequently accessed educational resources included; textbooks, e-books and e-libraries (35/148, 24%), updates/guidelines on surgical and other societies' websites (29/148, 19.5%), small group, interactive, online teaching sessions (26/148, 17.5%), pre-recorded teaching sessions (17/148, 11%), online papers/journals (15/148, 10%) and peer sharing, such as online forums and WhatsApp® groups (6/148, 4%).

The introduction of novel training methods was not without challenges. Technical issues for accessing online educational materials (106/418, 25%), small number of virtual or simulation training sessions (77/418, 18%), inappropriate timing for webinars (54/418, 13%), lack/

inability to have 'hands-on' training on simulated patients or simulators (45/418, 11%), difficulty to engage to maintain concentration during online sessions (31/418, 7%), lack of interaction during online sessions (30/418, 7%) were some of the problems faced. These led to the majority of responders being dissatisfied with these educational resources (254/608, 42%).

## 4.3. The proposed framework for recovery

According to the trainees who responded to our survey, ongoing online teaching sessions (92/307, 30%), prolonging training time (46/307, 15%), prioritising training and educational activities over service provision post-pandemic (36/307, 12%), increased use of simulation (29/307, 9%), mentorship by senior surgeons (28/307, 9%), proactive guidance from training authorities (24/307, 9%) and recommencing elective work (23/307, 7%), can help mitigate the negative impact of the pandemic on surgical training.

Although there appeared to be understanding that the negative impact on training was inevitable, there was apparent frustration for the perceived lack of response by training authorities in addressing this. Many responders highlighted the need for training authorities and trainers to communicate in order to formulate a concrete plan to prioritise training during the post-COVID-19 period so that trainees would be assisted to "make-up" for experience lost during the pandemic. Mentorship by senior doctors, use of simulation, e-learning methods and telemedicine were favoured.

The overall results of the survey provided evidence of widespread global disruption of all aspects of surgical training. Similar results have been reported in other studies [15, 25–33]. Alternative resources were introduced rapidly; however, trainees expressed some dissatisfaction with these.

Experience from previous pandemics has shown that disruption to training may be prolonged [34], so the development of a strategy for recommencing and maintaining training is of the utmost importance. Based on the results of the global survey conducted a four pillar framework is proposed:

**4.3.a. Four pillars.** *4.3.a. (i) Guidance from training stakeholders with trainee involvement.* The global survey identified lack of guidance from training stakeholders as one of the obstacles to training during the pandemic. While there was sufficient guidance in regards to service provision [17], there was paucity of directives for training. Several societies launched on-line educational platforms [35], however this was done in an uncoordinated manner, often resulting in duplication and sharing of conflicting information.

Training stakeholders need to improve communication both with each other and with trainees, coordinate educational activities and produce relevant guidelines. Timely communication between learners, hospital management, educators and training committees are vital. Standards to address training needs during a pandemic have been proposed and include prioritisation of training (over service provision), promotion of learner wellbeing, maximization of educational value and transparent communication [36].

*4.3.a. (ii) The role of senior surgeons/trainers in the prioritisation of training.* Hospitals worldwide should be encouraged to emphasise the importance of training alongside service provision. The hiatus of elective surgery during the COVID-19 crisis has created a significant backlog of patients. Under these circumstances surgeons may be apprehensive in providing training in the operating theatre due to time restrains and service provision commitments. Trainers should exhibit strong leadership [37] and be actively encouraged to train and mentor young surgeons both in and outside the operating theatre. Interventions to improve the efficiency of service provision should be developed to assist in the timely provision of appropriate surgical services to our patients after the pandemic. Virtual consultations, wider use of

"virtual" MDTs and telemedicine [38] might all contribute in this way. Mentorship was repeatedly mentioned in the survey and seems vital for effective and efficient surgical training.

*4.3.a. (iii) Access to alternative/new teaching methods.* Webinars, educational videos, e-libraries and simulation are popular amongst trainees [26, 38]; their use should be facilitated during and after the pandemic [35]. Hospitals should provide access to an adequate internet network, hardware and software and simulators. Simulation centres should consider expanding their working hours or find alternative methods to give trainees twenty-four-hour access. Lack of such access, has been identified previously as one of the barriers to simulation usage by trainees [39, 40]. In addition, simulation training is more effective in the presence of a trainer (instead of self-driven), therefore, appointing trainers for simulation sessions may accelerate the learning effect [40, 41].

**4.3.a. (iv) Addressing trainee anxiety.**

- **Recommencing elective activities**

Responders suggested that elective activities should be undertaken cautiously while maintaining patient and staff safety as a priority. Adequate Personal Protection Equipment (PPE) and avoidance of face-to-face interaction when possible seem imperatives, especially with high numbers of cases with COVID-19 in the community. The association between low number of cases and preservation of surgical training was apparent in the differences in responses between Europe and Australia / Asia. The significantly lesser impact on training in Australia and Asia, compared to Europe may well reflect differences in numbers of affected patients between continents during the survey period [42]

- **Training and career progression**

In addition to the pandemic related anxiety [43] surgical trainees experienced worry about career progression due to cancellation of exams [44] and training placements [45]. Annual review and recruitment processes had to be modified at very short notice, in order to comply with public health measures regulations [45]. Consideration must be given to assessment of competency for progression. Prolongation of training might be offered as a voluntary option to trainees. Hospitals should offer wellbeing sessions [46] for staff, to mitigate the adverse effects of the pandemics on mental health.

## 5. Presentation of the hypothesis and description of the proposal objectives

The recent COVID-19 outbreak demonstrated the vulnerability of healthcare systems in simultaneously managing education and patient care when a crisis occurs. It would appear that no health care service is immune; whether in an affluent country or not, public fund driven or insurance based. Politicians and healthcare leaders used reactive policies for dealing with rapidly changing situations that stemmed from lack of preparation. Taking a proactive approach rather than reactive attitude may minimise unintended effects such as the curtailment of surgical training. Fig 2 illustrates a proposal for policy development. It is apparent that trainers' involvement is paramount for the proposed framework's success. Therefore, their opinions need to be sought and a consensus should be reached on how to best deal with this unprecedented crisis in surgical training.

### 5.1. Objectives

5.1.a. Obtain the trainers' perspective on the framework developed from the global trainee survey. Document any additional ideas. This is expected to be achieved through semi-structured interviews (n = 10 surgical educationalists).

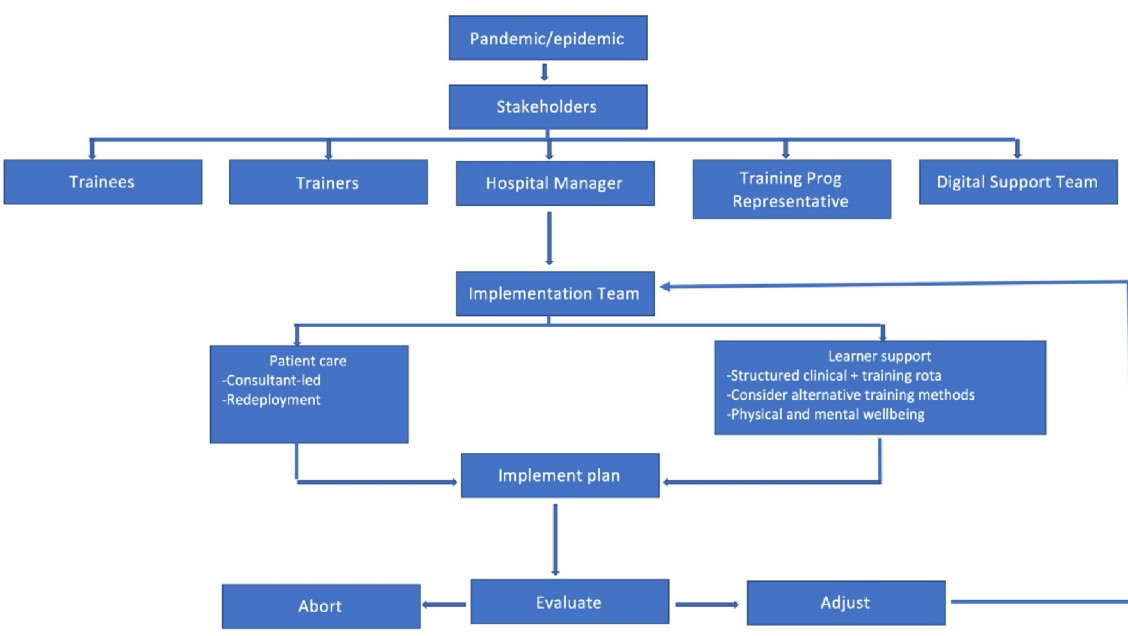

**Fig 2. Proposed framework for recovery.**

5.1.b. Obtain global consensus on a framework for recovery of surgical training with special focus on laparoscopic training. This will be achieved through a Delphi survey which would have been pre-validated.

5.1.c. Create a comprehensive guide on steps to be taken for surgical training recovery after and during the pandemic. This will reflect the desires and opinions of both trainees and trainers and will be pragmatic, allowing for the swift and straightforward implementation in the near future.

## 6. Methodology, inclusion and exclusion criteria, study design, data collection and statistical analysis

This will be a mixed qualitative, quantitative study, as it will encompass both semi-structured interviews and a global Delphi-style study.

The inclusion criteria for the responders of the semi-structured interviews and surveys are as follows:

i. Currently a practicing clinician, in a surgical unit performing both open and minimally invasive procedures.

ii. Has a formal role as a trainer and/or significant research portfolio related to medical education.

iii. Has demonstrated continuous commitment to surgical education for at least five years. This includes acting as faculty for courses, mentoring/training more junior trainees or having a formal role (e.g. lecturer, professor, surgical tutor, member of training committee–regional, national or international) in pre or postgraduate surgical training and/or education.

Exclusion criteria

i. Is a member of the steering committee of this project.

ii. Is not an expert surgeon (i.e. a resident or trainee) as this study aims to understand the opinion of experts.

The study will be performed in several stages

## 6.1. Stage 1: Preparing the semi-structured interviews

Based on the results of the global trainee survey, the steering committee (Ian Chetter, Marina Yiasemidou, Chandra Shekhar Biyani, Hector Guadalajara, Bijendra Patel) of this project is expected to reach a consensus on five open ended questions to be asked during the interviews. The process will involve each member of the steering committee independently preparing a list of 10 questions. This will be done after studying carefully the results of the global survey. The committee will then have a series of virtual meetings, until an absolute (100%) consensus is reached on which questions should be asked. The number of questions is not rigidly defined, therefore if the members of the steering group feel that questions need to be added or removed, this can be done after the necessary consensus is reached.

## 6.2. Stage 2: Delivering the semi-structured interviews

Once there is consensus on the open-ended questions, ten surgical educationalists will be selected for the semi-structured interviews. In order for these individuals to be identified, emails will be sent out and a 'call' will be put out on social media. Co-applicants on this project are not excluded as long as they are not part of the steering committee. If there is an abundance of volunteers, the steering committee will opt for a greater geographical spread (i.e. avoid two experts from the same country).

The interviews will then be delivered virtually by one of the members of the steering group, who will be at liberty to ask clarifying or 'follow-up' questions. The interviews will be audio recorded after relevant consent is obtained by the interviewee. These will then be transcribed verbatim.

The consent will be obtained in writing after the surgeons will read the participant information sheet (consent document and participant information sheet were both submitted and approved by the Hull York Medical School Ethics Committee).

## 6.3. Stage 3: Thematic analysis of the transcripts of the semi-structured interviews

The transcripts of the interviews, transcribed verbatim, will be analysed by two independent assessors (Marina Yiasemidou, Judith Long) who have previous experience with such analysis. The emerging themes will be presented to the steering committee of the project and used to inform a 5–10 question, Delphi style survey. Once again, the committee will meet virtually on several occasions until a >80% consensus is achieved for the questions suitable for inclusion in the Delphi-style survey.

## 6.4. Stage 4: Dissemination of Delphi-style survey and interim analysis

The co-applicants will be asked to identify five peers whom they will put forward as respondents for the Delphi-style survey. The recruits will have to match the same inclusion criteria as the educationalists answering the semi-structured interviews. We will aim to recruit fifty educationalists with the greatest geographical spread possible. After every cycle of the Delphi survey, an interim analysis of the results will be conducted. The process will be repeated until >80% agreement is achieved.

### 6.5. Data collection and analysis

**6.5.a. Semi-structured interviews–thematic analysis.**  The thematic analysis of the semi-structured interviews will be completed in the following steps: (i) familiarisation (ii) coding (iii) generating themes (iv) reviewing themes (v) naming themes (vi) write up of results [47]. Familiarisation will occur during the transcribing of the interviews. Coding will involve highlighting sections of the transcript and generating "codes" to describe their content. For example, if the transcript states, "I am not sure" the code can be "uncertainty". During the next step of generating themes, the codes created will be assessed for patterns amongst them. This process will generate themes; a broader concept than codes. On most occasions, several codes will be combined into a single theme. Following this, the themes will be reviewed by the two assessors (M.Y, J.L) to ensure that they are useful and accurate representations of the data.

This may result in themes being split up, combined, discarded or generated. This process will be ultimately validated by the steering committee. The themes developed will then need to be named in a succinct and precise way. The last step of the process will be the write up of the report, which will be prepared and delivered to the steering committee for careful scrutiny.

**6.5.b. Delphi-style survey.**  Data collection will be performed using a bespoke spreadsheet. Besides the answers of the survey, the demographics and educational experience of the responders will be collected. More specifically, the gender, years in practice, whether they have a formal trainer role, years spent as a trainer and specialty will be recorded.

The results from every Delphi-style survey cycle will be summated to assess for a consensus rate for every question. The number of questions within the Delphi style survey will be expected to increase and the context changed between cycles. Experts will be given the option to comment in a free text box and explain their answer for each question. This will enable the steering committee (acting as a facilitator for the Delphi style survey) to modify the questions of the next cycle in an accelerate achieving a consensus [48]. A dedicated statistician (Dr Zao, University of Hull, UK) will oversee and facilitate this process.

### 6.6. Strengths and limitations

The result of this study will be based on expert opinion which is the lower level of evidence, however the fact that project is run and addressed to an international and diverse audience, should add extra scientific value to the end result. Moreover, the experts/trainers are the ones who will lead the changes required in surgical training and this project will create an evidence based blueprint to guide this process.

### 6.7. Ethical approval and reporting guideline compliance

Approval for this study has been granted by Hull York Medical School Ethical Committee, Hull, United Kingdom (UK), on the 28[th] January 2021 –Reference number: 21 03.

The closest relevant guidelines suggested by the Enhancing the QUAlity and Transparency Of health Research (EQUATOR) network guidelines are Guideline for Reporting Evidence-based practice Educational interventions and Teaching (GREET) [49], and the reporting of suggested interventions will be reported based on these guidelines to the extent possible.

## 7. Work plan, broken down by task, investigators and timetable

The work plan and timetable for this project will be as follows:

### 7.1. Stage 1: Preparing the semi-structured interviews

This stage will be completed by the members of the steering committee.

### 7.2. Stage 2: Delivering the semi-structured interviews

Any of the co-applicants will be able to conduct the interviews. The overall process of preparing and delivering the semi-structured interviews is expected to take two months.

### 7.3. Stage 3: Thematic analysis of the transcripts of the semi-structured interviews

The validation of the survey will be completed by the steering committee and is expected to take two months.

### 7.4. Stage 4: Dissemination of Delphi-style survey and interim analysis

The dissemination will be completed by the co-applicants. This, the interim analysis and simultaneous data collection/analysis will be completed within five months. The data collection will be co-ordinated by M.Y and J.L.

### 7.5. Data collection and analysis

**7.5.a. Semi-structured interviews–thematic analysis.** The themes will be reviewed by the two assessors (M.Y, J.L), a report will be generated and presented to the steering committee. The entire process will be overseen and validated by the steering committee.

**7.5.b. Delphi-style survey.** The steering committee will be acting as a facilitator for the Delphi style survey and will collectively modify the questions of the next cycle in an attempt to accelerate the achievement of a consensus [48]. A dedicated statistician (University of Hull, UK) will be involved in the data collection and analysis from the beginning and has advised on the appropriateness of statistical methodology utilised in this project.

## 8. Description of the proposal's scientific and social interest

Surgical training was decimated during the COVID-19 pandemic. Anecdotal evidence indicates this to be particularly true for minimally invasive surgery; identified as an aerosol producing process and refrained from during the first wave of the pandemic. Surgical societies have been rather slow to react, enhancing the uncertainty amongst trainees. The paucity of relevant guidance should be addressed to assist surgeons (both trainees and trainers) during this difficult time. With the proposed project the authors are hoping to inform future guidelines by EAES for the recovery of surgical training during and post the COVID-19 pandemic.

As shown by the responses to our global survey, trainees are seeking mentorship from their trainers. Therefore, it is important to open channels of communication between trainees and trainers and involve both in the process of rebuilding surgical training. The first step for trainers' involvement is to listen to their opinions and ideas, which is what we intent to do with this study.

Previous pandemics and their aftermaths lasted for at least two years. The COVID-19 pandemic could follow a similar timeline, which would lead to trainee surgeons having significantly less experience than their predecessors. Such a prolonged training hiatus will lead to the deskilling of the workforce to such a great degree that would take time, resources and significant financial input to regain the lost skills. This will have a grave impact on health care systems struggling to cope not only with the pandemic itself but also the backlog of elective and semi-elective cases which will undoubtedly be created by the suspension of non emergency surgery during the COVID-19 crisis.

It is paramount that trainees and trainers collaborate so that concerns with training can be addressed with a unified front to proactively find solutions to maximise trainee opportunities

during these difficult times. Perhaps more importantly, surgical societies should prepare guidelines for the recovery of training after the pandemic. Acknowledging the great contribution of EAES to surgical training, we hope to use this project to generate evidence that can inform their future guidelines.

## Author Contributions

**Conceptualization:** Marina Yiasemidou, Peter Sedman, Damian Garcia-Olmo, Hector Guadalajara, Shekhar Chandra Biyani, Bijendra Patel, Ian Chetter.

**Data curation:** Marina Yiasemidou, Annabel Howitt.

**Formal analysis:** Marina Yiasemidou, Annabel Howitt, Judith Long.

**Funding acquisition:** Marina Yiasemidou, Judith Long.

**Investigation:** Marina Yiasemidou, Annabel Howitt, Ben Van Cleynenbreugel, Shekhar Chandra Biyani, Jonathan Robinson, Rafael Tourinho-Barbosa.

**Methodology:** Marina Yiasemidou, Annabel Howitt, Judith Long, Ben Van Cleynenbreugel, Shekhar Chandra Biyani, Athur Harikrishnan, Jonathan Robinson, Tiago Manuel Ribeiro de Oliveira, Rafael Sanchez-Salas, Rafael Tourinho-Barbosa.

**Project administration:** Marina Yiasemidou, Peter Sedman, Hector Guadalajara, Ben Van Cleynenbreugel, Ian Chetter.

**Resources:** Annabel Howitt, Judith Long, Ben Van Cleynenbreugel, Dhananjaya Sharma, Shekhar Chandra Biyani, Jonathan Robinson.

**Supervision:** Damian Garcia-Olmo, Hector Guadalajara, Dhananjaya Sharma, Shekhar Chandra Biyani, Bijendra Patel, Wayne Lam, Athur Harikrishnan, Gabriel Escalona Vivas, Ian Chetter.

**Validation:** Marina Yiasemidou, Judith Long, Shekhar Chandra Biyani, Wayne Lam, Juan Gómez Rivas, Jonathan Robinson, Tiago Manuel Ribeiro de Oliveira, Gabriel Escalona Vivas, Rafael Tourinho-Barbosa.

**Writing – original draft:** Marina Yiasemidou, Judith Long, Peter Sedman, Hector Guadalajara, Shekhar Chandra Biyani, Ian Chetter.

**Writing – review & editing:** Marina Yiasemidou, Annabel Howitt, Judith Long, Peter Sedman, Damian Garcia-Olmo, Hector Guadalajara, Ben Van Cleynenbreugel, Dhananjaya Sharma, Shekhar Chandra Biyani, Bijendra Patel, Wayne Lam, Athur Harikrishnan, Juan Gómez Rivas, Jonathan Robinson, Tiago Manuel Ribeiro de Oliveira, Gabriel Escalona Vivas, Rafael Sanchez-Salas, Rafael Tourinho-Barbosa, Ian Chetter.

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
