## [Decision Letter · Decision Letter 0]

17 May 2022

PONE-D-22-05245An international consensus for mitigation of the detrimental effects of the COVID-19 pandemic on laparoscopic trainingPLOS ONE

Dear Dr. Yiasemidou,

Thank you for submitting your manuscript to PLOS ONE. After careful consideration, we feel that it has merit but does not fully meet PLOS ONE’s publication criteria as it currently stands. Therefore, we invite you to submit a revised version of the manuscript that addresses the points raised during the review process.

We look forward to receiving your revised manuscript.

Kind regards,

Antonio Simone Laganà, M.D., Ph.D.

Academic Editor

PLOS ONE

Journal Requirements:

Additional Editor Comments (if provided):

The reviewers have expressed positive comments regarding your article, raising only few concerns. Considering this point, I invite authors to perform the required minor revisions.

Reviewers' comments:

Reviewer's Responses to Questions

**Comments to the Author**

1. Does the manuscript provide a valid rationale for the proposed study, with clearly identified and justified research questions?

Reviewer #1: Yes

Reviewer #2: Yes

2. Is the protocol technically sound and planned in a manner that will lead to a meaningful outcome and allow testing the stated hypotheses?

Reviewer #1: Yes

Reviewer #2: Yes

3. Is the methodology feasible and described in sufficient detail to allow the work to be replicable?

Reviewer #1: Yes

Reviewer #2: Yes

4. Have the authors described where all data underlying the findings will be made available when the study is complete?

Reviewer #1: Yes

Reviewer #2: No

5. Is the manuscript presented in an intelligible fashion and written in standard English?

Reviewer #1: Yes

Reviewer #2: Yes

6. Review Comments to the Author

You may also provide optional suggestions and comments to authors that they might find helpful in planning their study.

Reviewer #1: The Authors should be congratulated for their efforts in devising an international survey and consensus on factors contributing to disruption of surgical training during the Covid19 pandemic, and in producing a protocol for optimization of laparoscopic training. The primary study aim is to establish a Delphi consensus among trainers on how to recover lost training opportunities during and after the COVID-19 pandemic. The secondary study aim is to inform future society guidelines about the recovery of surgical training before and after the pandemic. The manuscript is well written and is of great interest to the global community of general surgeons. In my opinion, it may have a beneficial impact on the education of future trainees and a high potential for article citation.

Reviewer #2: I read with great interest the Manuscript titled “An international consensus for mitigation of the detrimental effects of the COVID-19 pandemic on laparoscopic training”, which falls within the aim of this Journal.

In my honest opinion, the topic is interesting enough to attract the readers’ attention. Methodology is accurate and conclusions are supported by the data analysis. Nevertheless, authors should clarify some point and improve the discussion citing relevant and novel key articles about the topic.

Authors should consider the following recommendations:

- Inclusion/exclusion criteria should be better clarified.

- The authors have not adequately highlighted the strengths and limitations of their study. I suggest clarifying these points

- Does this manuscript conform the Enhancing the QUAlity and Transparency Of health Research (EQUATOR) network guidelines? It would be mandatory to declare about this element.

- Was this study registered? I could not find any information about this point.

- According to novel pieces of evidence, 3-Dimensional versus 2-Dimensional Laparoscopy may have a significant on the quality of training. I invite the authors to discuss this element referring to: PMID: 27046747; PMID: 35398529.

- Limitations on training during the Covid-19 pandemic may have severely impacted the opportunity to learn basic clinical and surgical skills. A potential strategy for overcoming these limitation was offered by simulation activities, which allowed trainees to receive basic training in our discipline and prevented an additional “lockdown” of their learning and development of skills. I recommend to discuss these points (author may refer to: PMID: 34651559; PMID: 33682831).

7. PLOS authors have the option to publish the peer review history of their article (what does this mean?). If published, this will include your full peer review and any attached files.

Reviewer #1: **Yes: **Prof. Luigi Bonavina

Reviewer #2: **Yes: **Salvatore Insinga

---

## [Author Response · Author response to Decision Letter 0]

16 Jul 2022

Dear Antonio Simone Laganà, 

Thank you and your reviewers for taking the time to review our protocol titled: “An international consensus for mitigation of the detrimental effects of the COVID-19 pandemic on laparoscopic training”.

Please find below our point-to-point discussion regarding what was kindly raised by your reviewers:

Reviewer 1: ‘The Authors should be congratulated for their efforts in devising an international survey and consensus on factors contributing to disruption of surgical training during the Covid19 pandemic, and in producing a protocol for optimization of laparoscopic training. The primary study aim is to establish a Delphi consensus among trainers on how to recover lost training opportunities during and after the COVID-19 pandemic. The secondary study aim is to inform future society guidelines about the recovery of surgical training before and after the pandemic. The manuscript is well written and is of great interest to the global community of general surgeons. In my opinion, it may have a beneficial impact on the education of future trainees and a high potential for article citation.

Response: Thank you very much for your comments and encouragement 

Reviewer 2: Point 1. I read with great interest the Manuscript titled “An international consensus for mitigation of the detrimental effects of the COVID-19 pandemic on laparoscopic training”, which falls within the aim of this Journal.

In my honest opinion, the topic is interesting enough to attract the readers’ attention. Methodology is accurate and conclusions are supported by the data analysis. 

Response: Thank you for your kind comments.

Point 2: Nevertheless, authors should clarify some point and improve the discussion citing relevant and novel key articles about the topic.

Authors should consider the following recommendations:

- Inclusion/exclusion criteria should be better clarified.

Response: We have clarified these further in our manuscript. Thank you

Point 3: - The authors have not adequately highlighted the strengths and limitations of their study. I suggest clarifying these points

Response: We have clarified these further. Thank you

Point 4: - Does this manuscript conform the Enhancing the QUAlity and Transparency Of health Research (EQUATOR) network guidelines? It would be mandatory to declare about this element.

Response: Thank you for this suggestion. The closest relevant guidelines suggested by the Enhancing the QUAlity and Transparency Of health Research (EQUATOR) network guidelines are Guideline for Reporting Evidence-based practice Educational interventions and Teaching (GREET), and the reporting of suggested interventions will be reported based on these guidelines to the extent possible. This is now reflected in the manuscript. 

Point 5: - Was this study registered? I could not find any information about this point.

Response: We have not registered our study as we have not found a registration site specifically for medical education surveys, however if there is one we are not aware of, please suggest it to the authors and we would be happy to oblige. Having said that the authors believe the publication of the protocol in your esteemed journal to be serving an equal purpose as registration, as readers will be able to compare our protocol methodology and outcomes to the end result once the study is completed and published. Thank you.

Point 6 and 7: - According to novel pieces of evidence, 3-Dimensional versus 2-Dimensional Laparoscopy may have a significant on the quality of training. I invite the authors to discuss this element referring to: PMID: 27046747; PMID: 35398529.

- Limitations on training during the Covid-19 pandemic may have severely impacted the opportunity to learn basic clinical and surgical skills. A potential strategy for overcoming these limitation was offered by simulation activities, which allowed trainees to receive basic training in our discipline and prevented an additional “lockdown” of their learning and development of skills. I recommend to discuss these points (author may refer to: PMID: 34651559; PMID: 33682831).

Response: We are very appreciative of the points made here. Simulation was one of the mitigation measures suggested in the trainee survey conducted by our group. However, we are now trying to establish the unbiased view of trainers, whilst the authors may agree on the merit of certain teaching methods, explicitly mentioning them in the study’s protocol may be considered leading.

Thank you 

Kind regards,

Marina Yiasemidou, MBBS, MRCS, MSc, PhD

NIHR Clinical Lecturer, University of Hull, UK

---

## [Editor Report · Decision Letter 1]

20 Jul 2022

An international consensus for mitigation of the detrimental effects of the COVID-19 pandemic on laparoscopic training

PONE-D-22-05245R1

Dear Dr. Yiasemidou,

We’re pleased to inform you that your manuscript has been judged scientifically suitable for publication and will be formally accepted for publication once it meets all outstanding technical requirements.

Kind regards,

Antonio Simone Laganà, M.D., Ph.D.

Academic Editor

PLOS ONE

Additional Editor Comments (optional):

I carefully evaluated the revised version of this manuscript.

Authors have performed the required changes, improving significantly the quality of the paper.
---

## [Editor Report · Acceptance letter]

29 Jul 2022

PONE-D-22-05245R1 

An international consensus for mitigation of the detrimental effects of the COVID-19 pandemic on laparoscopic training 

Dear Dr. Yiasemidou:

I'm pleased to inform you that your manuscript has been deemed suitable for publication in PLOS ONE. Congratulations! Your manuscript is now with our production department. 

Kind regards, 

on behalf of

Dr. Antonio Simone Laganà 

Academic Editor

PLOS ONE